# What Are the Psycho-Social and Information Needs of Adolescents and Young Adults Cancer Care Consumers with Intellectual Disability? A Systematic Review of Evidence with Recommendations for Future Research and Practice

**DOI:** 10.3390/children8121118

**Published:** 2021-12-02

**Authors:** Éidín Ní Shé, Fiona E. J. McDonald, Laurel Mimmo, Xiomara Skrabal Ross, Bronwyn Newman, Pandora Patterson, Reema Harrison

**Affiliations:** 1School of Population Health, University of New South Wales, Sydney, NSW 2052, Australia; 2Centre for Health Systems and Safety, Australian Institute of Health Innovation, Macquarie University, Sydney, NSW 2190, Australia; bronwyn.newman@mq.edu.au (B.N.); reema.harrison@mq.edu.au (R.H.); 3Canteen Australia, Sydney, NSW 2042, Australia; fiona.mcdonald@canteen.org.au (F.E.J.M.); xiomara.skrabal@canteen.org.au (X.S.R.); pandora.patterson@canteen.org.au (P.P.); 4Faculty of Medicine and Health, The University of Sydney, Sydney, NSW 2052, Australia; 5School of Women’s and Children’s Health, Faculty of Medicine, University of New South Wales, Sydney, NSW 2052, Australia; lmimmo@unsw.edu.au; 6Clinical Governance Unit, The Sydney Children’s Hospitals Network, Westmead, NSW 2145, Australia

**Keywords:** adolescents and young adults, intellectual disability, cancer care, psychosocial and information support, health, transitions, systematic review

## Abstract

People with intellectual disability have unmet health needs and experience health inequalities. There is limited literature regarding cancer care for children, adolescents, and young adults (AYA) with intellectual disability despite rising cancer incidence rates in this population. This systematic review aimed to identify the psycho-social and information support needs of AYA cancer care consumers with intellectual disability to generate recommendations for future research and cancer care service delivery enhancement. We searched eight databases yielding 798 articles. Following abstract and full-text review, we identified 12 articles meeting our inclusion criteria. Our three themes related to communication and accessible information; supports and system navigation, cancer service provider training, and reasonable adjustments. There was a lack of user-friendly, accessible information about cancer and screening programs available. Both paid and family carers are critical in accessing cancer supports, services, and screening programs for AYA with intellectual disability. Ongoing training should be provided to healthcare professionals regarding the importance of care screening for AYAs with intellectual disability. This review recommends that AYA with intellectual disability and their family carers be involved in developing tailored cancer services. This should focus on enabling inclusive screening programs, accessible consent, and challenging the enduring paternalism of support services via training and appropriate communication tools.

## 1. Introduction

Adolescents and young adults (AYA) who have been diagnosed with cancer have been recognised as a distinct population since the early 1990s [1] and it is estimated that globally there are over 1,000,000 new diagnoses each year [2]. While no global estimates exist, many more AYAs will have a sibling or parent diagnosed with cancer. Identified impacts for AYAs diagnosed with cancer include complex medical, psychosocial, emotional, and fertility concerns, with detrimental effects on quality of life, education, as well as physical and social development [3,4,5,6]. These impacts can extend for years post treatment during the survivorship phase [2,3,5,7,8]. While AYA siblings of cancer patients do not experience the physical aspects of cancer diagnosis and treatment, they can experience similar disruptions to their family routines alongside worries about their sibling, which can lead to increased distress and emotional and behavioural problems [7,8]. For many AYAs whose parents are diagnosed with cancer, the experience can be disruptive and detrimental with reports of very high levels of psychological distress and associated unmet needs [9]. As a group, AYAs impacted by cancer experience have considerable information and support needs [9,10]. Information needs include wanting to understand what is happening with treatment in age-appropriate language and feeling listened to, while support needs include knowing how to manage changes in relationships with friends and family, disruption to normal home and education routines, and feelings of isolation [8,9,10,11,12]. These considerable disruptions can lead to increased mental health concerns often evidenced by higher levels of psychological distress and reduced well-being.

While the support needs of marginalised sub-groups of cancer patients have been identified as needing particular attention if equitable treatment, support, and care is to be provided to all cancer patients [13,14], the limited research to date involving disadvantaged AYAs impacted by cancer has focused on gender, poverty [15], residential locality, race, and insurance status [15,16]. To date, it appears there has been little to no attention paid to the experience of young people with intellectual disability who also have either a personal or familial cancer experience, and in particular no attention has been given to their information or psychosocial support needs.

It is estimated that globally approximately 1% of the population have an intellectual disability and that this is higher amongst children and adolescents [17]. Intellectual disability is characterised by impairments in intellectual functioning (typically an IQ of below 70) and adaptive functioning, including skills required for independent daily living, with onset during the developmental period [18]. While intellectual disability is the preferred term in Australia, Ireland, Canada, New Zealand, and the United States, the term ‘learning disability’ is commonly used in the United Kingdom [19]. As per Cluley [19], intellectual disability is the preferred term internationally and is used in this paper.

A recent review of admissions to Sydney Children’s Hospitals Network in 2017 for children aged 0–18 years found that 13.9% of admissions greater than 23 h involved a child or AYA with intellectual disability [20]. In addition, 8% of cancer-related admissions in the sample were for children with intellectual disability [21,22,23,24]. It is well-established that people with intellectual disability face barriers when accessing health care, including limited organisation knowledge about intellectual disability, inhibitive staff perceptions, and problems with communication [25]. A recent systematic review exploring the psychosocial experiences of people with intellectual disability and chronic illness found there are difficulties in communicating with care teams and understanding the illness, leading to feelings of uncertainty, confusion, and distress [23]. AYA who have intellectual disability and experience cancer in their lives, either through their own or a family member’s diagnosis, are likely to have specific support and information needs. It is vital that these needs are recognised to ensure appropriate supports are available.

## 2. Purpose and Question

### 2.1. Study Purpose

The aims of this review were to systematically identify and appraise the existing evidence of the psycho-social and information support needs of AYA with intellectual disability who access cancer services, as a patient, sibling and/or other family member of someone diagnosed with cancer. Specifically, our objectives were:

1. Identify the current evidence regarding information and psychosocial support needs of adolescents and young people aged 15–25 with intellectual disability.

2. Thematic analysis and narrative synthesis of the information and psychosocial support needs of AYA cancer care consumers with intellectual disability.

3. Generate recommendations for future research and cancer care service delivery enhancement.

### 2.2. Study Question

What are the psycho-social and information support needs of AYA cancer care consumers with intellectual disability, as a patient, sibling, and/or other family member?

## 3. Methods

The review protocol was registered with PROSPERO, which is an international registry for systematic reviews (CRD42021224101) [26]. We conducted a systematic search and narrative synthesis according to Green et al. [27].

### 3.1. Search Strategy

To ensure a search strategy that was both sensitive and specific, a comprehensive search methodology to identify both published and grey (e.g., policy reports, national/international guideline documents, etc.) literature was developed and executed through routine scientific database searches and grey literature retrieval. Key search terms were agreed by the team in consultation with a librarian (Appendix A). The search period was from the 1st of January to 30th of March 2021. We systematically searched the following academic databases: Medline; Embase; Emcare; PsycINFO; CINAHL; Web of Science; Scopus; and the Cochrane library. Two grey literature repositories were also searched: Open Grey, Open Doar and eight identified websites (refer to Appendix A). Seven additional journals were also identified for hand searching (refer to Appendix A).

### 3.2. Search and Selection Process

Identified studies were screened for eligibility using Covidence (www.covidence.org, accessed on 1 February 2021). All studies had their title and abstracts screened, and potentially relevant studies had their full text reviewed to determine eligibility. Meeting fortnightly, disagreements were resolved by discussion between at least three authors (É.N.S., F.M.D., L.M.).

### 3.3. Eligibility Criteria

Studies had to meet the following criteria to be included:

#### 3.3.1. Inclusion Criteria

Types of Studies

Published from 1 January 2000 [28] until 31 January 2021.English language.Studies involving AYAs aged 15–25 years with intellectual disability diagnosed in individuals less than 18 years who demonstrate permanent impairments to learning, thinking and reasoning, and social functioning [18].Current or past cancer care consumer, including as a patient, as a sibling, or other family member of someone accessing cancer care services in any cancer care setting and in any country.Parent/carers, siblings or health professionals acted as proxies for AYA with intellectual disability.

#### 3.3.2. Exclusion Criteria

Studies that do not report specific data related to AYAs aged 15–25 years of age.Studies involving AYA without intellectual disability only.Studies of AYA with chronic and/or complex conditions but do not have intellectual disability. For example, dyslexia, hearing loss/speech delay, reading difficulty, physical disability, Autism/Autistic Spectrum Disorder without intellectual disability [29], and mental health disorder.Prevalence studies.

#### 3.3.3. Outcomes

Data regarding information and/or psychosocial needs; defined as the emotional and well-being needs beyond physiological needs, of AYA cancer services and supports for consumers with intellectual disability.

### 3.4. Data Extraction

A narrative synthesis of extracted data was conducted because of the substantial methodological heterogeneity between studies to identify patterns within the literature [30,31]. The findings of included publications were reviewed using the narrative synthesis approach, based on the study objectives, to elicit key themes outlined in the papers [31]. Three team members (É.N.S., F.M.D., L.M.) extracted the following information for each study: authors, publication year, country, study design, cancer type, intellectual disability, sample size and psychological interventions. Thematic analysis was undertaken on the papers and initial coding was developed following inductive analysis [32]. Via weekly meetings, data extraction and analysis were completed by É.N.S., F.M.D., and L.M. and reviewed and agreed upon by all authors.

### 3.5. Quality Appraisal

The quality of the included studies was assessed to determine the robustness of the results and how much weight should be given to each study when interpreting patterns in the literature overall. One author (ÉNS) used the 13-item Quality Assessment Tool for Diverse Studies (QuADS) as the appraisal tool. QuADS was selected as a validated instrument that is best suited to appraising a heterogenous group of studies or mixed and/or multi-methods studies within systematic reviews [33]. Study quality was not an inclusion/exclusion criterion for inclusion in the review but is presented in the published summary table.

## 4. Results

### 4.1. Study Selection

A total of 826 studies were identified from the literature, including seven journal articles obtained from handsearching. After duplicates were removed (*n* = 28), 798 studies were screened for inclusion. A total of 748 studies were excluded after title and abstract screening and 38 were excluded after full text evaluation. A total of 12 studies were included in this review. See Figure 1 for the PRISMA flowchart.

### 4.2. Study Characteristics

The 12 included studies were published between 2007–2020 [22,34] and their characteristics are presented in Table 1. Nine studies were from the United Kingdom [21,22,23,24,34,35,36,37,38] two studies from Canada [39,40] and one from Australia [41]. Three studies combined a literature review with expert opinion and reflections and/or contributions from relevant stakeholders [38,40,41]. Two studies were systematic reviews, one published in 2020 [34] and the second in 2007 [22]. A study from Canada reviewed health administrative databases and registries to review cancer screening utilisation by women with intellectual disability (*n* = 17,777) focused on cervical and breast cancers compared with a 20% sample from a database of women with an intellectual disability (*n* = 1,440,962) [39]. Two surveys were undertaken. One study captured oncology nurses’ views of caring for people with intellectual disability [23]. The second was a survey of care staff on how they engaged in cancer prevention and health promotion activities with people with intellectual disability [21]. There was one evaluation from 2007 of a cancer information pack for people with an intellectual disability to support enhanced communication and understanding [36]. Three studies were qualitative. One undertook interviews with six people with an intellectual disability and twelve from their support network to capture their experiences of cancer [24]. The second qualitative paper involved four people with intellectual disabilities affected by cancer to inform the development of a research agenda [35]. The third paper used an ethnographic approach involving 13 interviews with family and paid carers supported by focused observations on women with intellectual disabilities attending breast awareness and breast screening programs [37]. Out of the 12 studies extracted, nine papers either did not provide age breakdown of participants or those involved were over the age of 25. This was an initial exclusion criterion but following discussions it was agreed to retain the nine studies due to their relevance. Three papers included and identified adolescents or young adults between the ages of 18 and 25 with an intellectual disability, along with older participants [22,35,39]. Two of the identified papers mentioned children impacted by familial cancer [31,37]. This was in the context of proposed future research [37] and a study with participants who were bereaved due to familial cancer [31]. Due to the low number of papers, the scope was broadened to include screening papers, and it is important to note that these studies did not focus on people diagnosed with cancer.

Following thematic analysis, three themes were agreed upon linked to the psycho-social and information support needs of AYA with intellectual disability:Communication and Accessible InformationSupports and System NavigationCancer Service Provider Training and Reasonable Adjustments

### 4.3. Communication and Accessible Information

The lack of accessible communication for AYAs with intellectual disability about cancer, either their own or about a family members diagnosis, was a major finding in all of papers extracted. There was a lack of user friendly, accessible information about cancer and screening in particular [21,34]. A recent systematic review published in 2020 focused on capturing the attitudes and opinions of people with learning disabilities, family carers, and paid care workers towards national cancer screening programmes [34]. A total of eleven papers met their study criteria and were all related to cervical and breast screening. The review found the need to shift communication and information sharing to women accessing screening programs to become more person-centred. In particular, letters sent to women inviting them to attend screening programs were not accessible, and the review recommended the need for modifications to the invitations process to include visual and alternative communications aids or electronic recordings [34]. This point was also made in a 2015 study capturing the views and experiences of paid- and family-carers when supporting women with intellectual disabilities through breast screening [37]. The study found that how carers approached and explained the screening following the arrival of an invitation letter had a direct impact on the woman’s final decision about participating in the procedure [37]. A paper evaluating a communication tool developed for people with intellectual disabilities found that it enabled users to communicate and exercise choice by developing their own understanding [36].

### 4.4. Supports and System Navigation

The role of both paid and family carers was addressed as key gatekeepers and enablers in accessing cancer supports, services and screening programs for AYA with intellectual disability [23,24,34,35]. A study from Canada that reviewed the literature and included the expert insights of the authors stressed the need of women with intellectual disability and their caregivers for more support and guidance during reproductive health care to enable their involvement [40,41,42]. A second study from Canada compared the attendance rates of cervical and breast cancer screening of women from the general population with women with intellectual and developmental disabilities [39]. The study found the proportion of women with intellectual and developmental disabilities who are not screened for cervical cancer was nearly twice that in the women without a disability and 1.5 times that for mammography [39].

The role of family caregivers and paid staff was noted as crucial in providing information on cervical and breast cancer screening, in supporting the person during the procedure and reporting any potential symptoms of cancer to health professionals. Developing interventions on information and training to support caregivers and staff in this role was recommended [39]. One paper from the UK argued that the role of family carers as gatekeepers was a taboo topic [35]. For people with intellectual disability, an underlying cause of negative experiences was that they were frequently bypassed when information was being provided [35]. This resulted in the family carer having control over the level of information to share with the person [35]. There was also evidence of paternalistic attitudes outlined in the paper with gatekeepers reporting that they felt discussing experiences of cancer would be too upsetting [35]. The bypassing of people with intellectual disability within cancer consultations and being excluded from conversations about their care and treatment related decisions was noted in another UK study [24]. Caregivers (family and paid) were being relied upon by healthcare staff to facilitate communication and understanding and supplement health care professional knowledge. The paper outlined the need to enable increased empowerment and involvement of people with intellectual disability [24].

A recent systematic review from 2020 outlined that some family carers tried their best to balance their decision about supporting cancer screening and the likelihood of distress against the benefit to the person with intellectual disability [34]. If the family carer and/or paid care workers had an unfavourable opinion towards cancer screening programs this was likely to impact on the person with intellectual disability resulting in them feeling anxious and reluctant to partake [34]. The review recommends the need for proactive, person-centred cancer screening invitations, which do not focus on health literacy and family or paid care workers, but rather empower the person with intellectual disability to make an informed decision on whether to attend a screening program [34].

A review from 2017 focused on the literature on psychosocial support needs in the complex care of children with both life-limiting conditions and intellectual disability [41]. As so few papers were identified in their review, the authors, who worked as clinicians in a tertiary children’s hospital in Brisbane, Australia, presented their expert group recommendations. They report that research is needed in the areas of symptom management and care coordination, communication and decision making, psychosocial and bereavement support, and education/training for a child with a life-limiting condition and intellectual disability. Children’s understanding of death and dying of their close families, particularly for children with intellectual disability, was another suggested focus of further research [41]. One paper from 2010 reported specifically on the process of developing an advisory forum’ of people with intellectual disability affected by cancer [35]. The paper involved conversations with four people with an intellectual disability, one who had experienced cancer and three who were the child and/or close relative of a person who had died from cancer. The paper stressed the need to challenge the enduring paternalism of services and opening of a dialogue about how best to support people with intellectual disability to engage in their cancer care [35].

### 4.5. Cancer Service Provider Training and Reasonable Adjustments

Ensuring ongoing training of healthcare staff was a central finding in the literature focused on a number of key areas [21,34,38,40]. One such area noted from research in cancer screening is that training should be provided to healthcare professionals on the importance of health screening for people with intellectual disability and on how to support their patients in understanding and consenting to a cancer screening procedure [23]. A 2015 paper from the UK explored the perceptions of oncology nurses regarding the provision of cancer care for patients with intellectual disability [23]. Interviews with the nurses found a perception that looking after a cancer patient with intellectual disability would be more time consuming and complex. However, the research also found that previous experiences and increased training of caring for people with intellectual disability worked as an influencing factor against negative perceptions [23]. The authors recommend that training and information must be provided to professionals on the importance of health screening for people with intellectual disability, and this should support their understanding of and consent to cancer screening programs [21]. Training for social care staff in order to raise knowledge and awareness was highlighted in a survey with 40 carers [21]. A systematic review stressed that training needs should be focused on professional practice barriers including the need for enabling multidisciplinary working to support people with intellectual disability [34].

Screening programs in particular should implement reasonable adjustments to reduce anxiety and improve the experience [23,38] of people with intellectual disability. Any adjustments developed should involve the input of AYA with intellectual disability and their families [23,38]. An Australian paper incorporated the expert opinion of clinicians working in a children’s acute hospital on the challenges of providing paediatric palliative care to children with intellectual disability [41]. Throughout the paper, the clinicians stressed the importance of family-centred approaches, noting the need for time to enable shared decision making focused on growing trust and relationships between children, families, and healthcare professionals [41]. A scoping study focused on evidence and gaps in breast cancer support for women with intellectual disability included a consultation with relevant stakeholders in the UK city of Sheffield [38]. The work highlighted a dearth of research and practice guidelines to support women with intellectual disability who are diagnosed with cancer. Two other studies that noted this gap were literature reviews. One undertaken in 2007 focused on the evidence around the need of people with intellectual disabilities for palliative care [22]. The review found an almost total absence of insights from people with intellectual disability themselves to develop guidelines.

## 5. Discussion

To our knowledge this is the first systematic review to identify and synthesise the evidence on psychosocial support and information needs for adolescents and young adult cancer consumers who have intellectual disability. Twelve papers met the inclusion criteria, of which two papers included family members of cancer patients who have intellectual disability. Our review found a large cohort of the academic literature focused on prevalence studies which were excluded. One of the difficulties with identifying papers that included AYAs was that very few papers provided detailed demographic information that is typically found, such as the age of the participants. There was also limited information reported on the severity of intellectual disability and the appropriateness of resources or services for those with severe or profound intellectual disability.

Women’s health and cancer screening programs specifically, was the focus of the twelve papers, and we note the gap around men’s cancer screening programs. While important for positive cancer outcomes, screening programs are targeted at the general population, and therefore the inclusion of these papers meant including papers that were not focused on people impacted by a cancer diagnosis. Most of the papers were qualitative, with small sample sizes included. The papers highlight that for people with intellectual disability including AYAs with intellectual disability, how information is received impacts on their screening experience. There is an urgent need to develop accessible information about screening programs using appropriate methods, such as easily read information and videos. Making available information and training on the importance of screening is also a clear learning from this systematic review targeting both family and paid carers. There is also a significant gap in the literature on the role of primary and community care towards disease identification, providing the space for understanding and accessible education for AYA with an intellectual disability [42,43].

A significant finding from this systematic review was the lack of evidence on what were major support needs of cancer care consumers with intellectual disability [22,24,38,41]. Adolescence and young adulthood represent an age when people start to become responsible for their own health decisions, and for females in particular this is when cancer screening is initiated. For AYA with intellectual disability, the transition from child to adult care services can be particularly challenging and overwhelming [44]. A negative experience at this age has the potential to have a negative influence on screening throughout life, with the reverse also being true. We found no literature that specifically examined the emotional, social, or psychological experiences of AYA with intellectual disability impacted directly by a cancer diagnosis, nor on the experiences of AYA with intellectual disability who have a family member, such as a sibling or a parent, diagnosed with cancer. For AYA without an intellectual disability who are impacted by parental cancer, effective communication within the family and with parents especially is their greatest unmet need and a predictor for positive outcomes [45]. It is anticipated that AYA with intellectual disability who rely on family members for caring and support may experience considerable distress and disruption if someone in their family is diagnosed with cancer, and the family care givers may be unable to continue to provide the same level of support.

People with intellectual disability face inequities when accessing health and social care services [46]. It remains encouraging to note that the recent academic literature has started to question the lack of focus on the experience of cancer care and survivorship of people with intellectual disability [47,48,49,50]. However, more focused attention is needed. A recent scoping review focused on the context of cancer care in the United States recommends the urgent need for improving caregiver support, collaboration among health care providers, and ethical approaches to information disclosure for people with intellectual disability as well as the establishment of pathways of care [50]. Healthcare staff concerns around having the time and resources to deliver care required for people with intellectual disability were prominent in the included studies. The literature regarding healthcare experiences for children and adults with intellectual disability reported poor experiences. In a 2014 systematic review of poor hospital experiences for adults with intellectual disability, healthcare staff identified limited knowledge and a lack of information and training in caring for people with intellectual disability [51]. For hospitalised children and young people with intellectual disability, healthcare staff assumptions and lack of knowledge about the child with intellectual disability results in an overreliance on parents and carers to meet their child’s care needs [52,53]. Additionally, AYAs with intellectual disability who have a family member diagnosed with cancer are likely to be excluded from conversations with HCPs completely.

It is clear in the literature that adolescent and young adult oncology is now a well-established sub-specialty, distinguishable from paediatric and adult oncology in terms of incidence [52,53]. Their specific psychosocial characteristics and health service needs has been outlined in the academic literature [54,55,56]. Advances have occurred in embedding standards of care, screening for psychosocial risk and supporting the involvement of AYA in their care decisions [55,56,57]. Additionally, in some health systems, specialised services have been developed for AYA with cancer [3,58,59,60]. However, this systematic review notes that AYA with intellectual disability experience exclusions in accessing patient centred and inclusive health services whether they are the patient or a family member of the patient. This further marginalises this group and contributes to the health inequities experienced by children and AYA with intellectual disability [18]. Our review noted very little information was available on their perspectives. While there is a need for hospital-based staff to receive training and support in improving care for AYAs with intellectual disability, there is also a role for the community sector. In some instances, provision of appropriate communication tools and support in using them could be provided by community organisations who have the appropriate expertise. There is also the possibility for national community organisations to provide additional support to these vulnerable young people and their families, and to facilitate peer support.

### 5.1. Study Limitations

There were limitations to this review. Only three papers provided a specific age breakdown of participants that fit the adolescent and young adult criteria of 15–25 years and none focused on AYAs only. Articles were restricted to only those in the English language. However, the search terms for this review were purposely broad. There was a lack of continuity across studies in type of data, measures, and study design within the extraction. There was also inconsistent reporting on the ages or cancer diagnosis of participants with intellectual disability, in the papers, something that future papers should provide as a minimum. No evidence was available on the experiences of an AYA with severe and profound intellectual disability. However, a few papers reported on the severity of intellectual disability. Additional data and breakdowns on age would allow for additional analysis across studies. It is possible that relevant articles may have been missed. This risk was minimized by seeking the input of a research librarian to design the search string. We are confident that together with hand searching we have identified all relevant studies. Despite limitations, this systematic review provides valuable insights into the psychosocial needs of AYA with an intellectual disability impacted by cancer.

### 5.2. Implications for Practice

There is an urgent need to shift focus away from incidence studies of levels of cancer in people with intellectual disability. There is a clear case to support the need for professional training in supporting AYA with intellectual disability and developing appropriate information resources. Further research needs to enable the involvement of AYA with intellectual disability in shaping research priorities and describing the impact of cancer on them [26,60]. Consideration is also needed to better understand how to support AYAs with severe and profound intellectual disability, how to support AYA with intellectual disability who are siblings or children of cancer patients, including during end-of-life care, death and bereavement, how to support families and caregivers to provide support to these AYAs, and the role of primary and community care. Service providers and researchers need to collect information about intellectual disability and consider how the voices of those with intellectual disability can be heard. Following this, support services can be further tailored to meet their unique needs.

## 6. Conclusions

The needs of AYA with intellectual disability who are also impacted by cancer are not well understood. Research to date has focused on prevalence, with limited research also examining screening, information, and caregiver needs. To provide the necessary support to this group of young people, research is urgently needed that involves them, to understand their support and psychosocial needs.

## Figures and Tables

**Figure 1 children-08-01118-f001:**
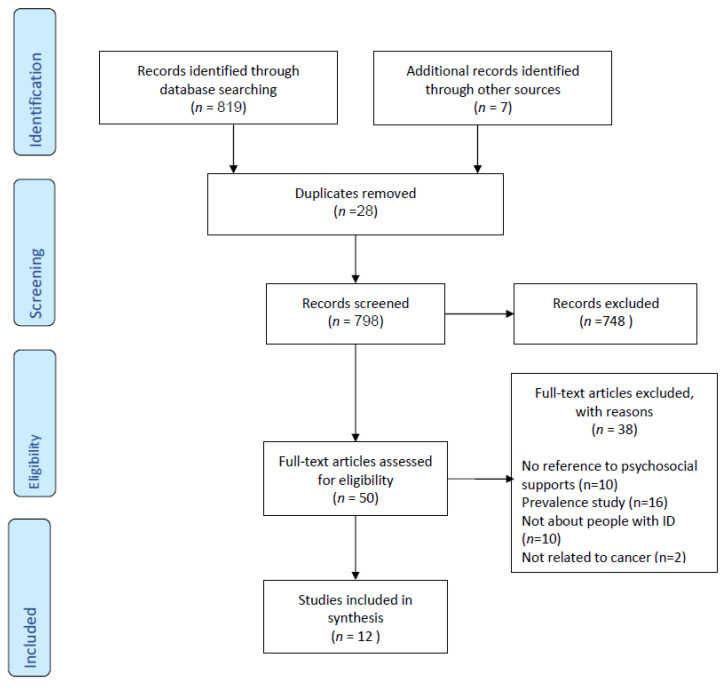
PRISMA Flowchart.

**Table 1 children-08-01118-t001:** Characteristics of reviewed studies.

Author & Country	Year	Study Objective	Methods	Cancer Diagnosis	Description of ID	AYA Mentioned Specifically [12,13,14,15,16,17,18,19,20,21,22,23,24,25]	Psychosocial Support Needs Identified	Quality Appraisal [33]
Flynn et al. [24] UK	2016	Present an account of cancer related experiences of people with ID & to generate a grounded theory with relevance to both research and practice.	Qualitative -Interviews 6 people with ID and Cancer and 12 from their support network	Plasma, Cytoma, Bowel Testicular, Stomach Breast Ovarian and Lung	No specifics -those included had to have a mild ID (IQ = 55–70)	No-Youngest person with ID was 34	People with ID were often overlooked within cancer consultations and excluded from conversations about their care and treatment related decisions. Caregivers (family and paid) were relied upon to facilitate communication and understanding and supplement health care professional knowledge. Need to support increased empowerment and involvement of people with ID. Enhanced patient centred skills for healthcare staff.	29/39
Collins et al. [38] UK	2014	Capture relevant research and identify gaps in evidence about breast cancer & ID	Scoping study that included consultation with stakeholders (n = 26) from one city Sheffield in the UK	Breast Cancer/Breast Screening	A local advocacy group for people with ID were included-three women with ID were interviewed together	Not specified	Paper highlighted a dearth of research and practice guidelines on the information and support needs of women with ID across the breast cancer pathway. Recommend further research needed to develop appropriate protocols, strategies and interventions in order to address these gaps.	30/39
Byrnes et al. [34] UK	2020	Reporting the attitudes and opinions of People with a Learning Disability, family carers, and paid care workers towards national cancer screening programmes amongst UK populations	Systematic review of 11 papers related to cervical and breast screening. No papers were related to colorectal cancer screening.	Cervical and breast screening	Not specified	Not specified	Synthesis highlights four areas of significance: [1] supporting women with a learning disability (WwLD) to attend screening, [2] WwLD’s awareness of screening and their psychophysical experiences, [3] professional practice barriers including the need for multidisciplinary working and an understanding of the needs of WwLD, and [4] approaches to improve the uptake of cervical and breast cancer screening. The synthesis highlights the significance of WwLD having support to understand the importance of screening to be able to make an informed choice about attending.	34/39
Flynn et al. [42] UK	2015	Investigate the previously unexplored perceptions of oncology nurses regarding the provision of cancer care for patients with and without an ID	Survey with 83 oncology nurses using 4 vignettes on perceptions of caring for patients with and without an ID were measured, alongside potentially confounding information about participant demographic characteristics and perceived stress	Oncology-details not specified	Not specified	Not specified	Perception that providing cancer care to person with intellectual disability will be more difficult than that for a person without intellectual disability. ‘however previous experience and increased knowledge working with this specific patient group acts as a protective factor against negative effects.’ Authors recommend interventions that will increase staff awareness and knowledge of care needs of people with intellectual disability to ‘ reduce anxiety and improve the perceptions and attitudes of oncology nurses when caring for this group of patients	22/39
Forbat et al. [35] UK	2010	To describe ‘the process of developing an advisory forum’ of people with intellectual disabilities affected by cancer.	Individuals who consented to their involvement were engaged in qualitative conversations about their experiences of cancer.’ Using an accessible book describing cancer treatment for people with intellectual disability with breast cancer/lymphoma.	Breast Cancer and Lymphoma	4 participants with ID- One participant had cancer. All participants were the child and/or close relative of someone who had and/or died from cancer.	22–53 years. 1 female was 22 years old the other 3 participants were over 40 years of age.	Gatekeepers’ treatment of cancer Seen as a taboo topic. The need to approach people via gatekeepers highlighted paternalistic attitudes that prevented people from becoming involved. Gatekeepers reported that they felt discussing experiences of cancer would be too upsetting for the service users Challenging the enduring paternalism of services, and opening dialogue about how best to support people with intellectual disabilities to engage in their care, must be a first step. Research topics should focus on communication between healthcare professionals and people with intellectual disabilities; effective supports for family members, staff and people with intellectual disabilities when someone gets cancer; and incidence and prevalence profiles.	32/39
Gilbert et al. [36] UK	2007	Evaluate the ‘Living with cancer pack’. for people with intellectual disabilities to help them understand cancer and communicate their needs-	The evaluation strategy involved obtaining data from four different sources to evaluate the effectiveness of the ‘Living with cancer pack’. 1. Mapping dissemination of materials 2. Focus groups with people with intellectual disabilities 3. Postal questionnaire to organisations that received a pack 4. Follow up telephone interviews with the respondent to the postal questionnaire	Not specified	Not specified	Not specified	Evaluation suggests that the ‘Living with cancer’ pack provides an important tool to supporting people with intellectual disabilities in both ‘health promotion’ activities and ‘the cancer journey’. Focus groups with people with intellectual disability highlighted another key feature, that of communication and exercising choice. These materials can play a positive role in facilitating people with intellectual disability to develop a language about their bodies enabling them to express feelings or concerns about their health. In addition, these materials could prove invaluable in helping people with intellectual disability explore health choices and issues related to end-of life care.	29/39
Tuffrey-Wijne et al. [22] UK	2007	To review the evidence around the current need of people with intellectual disabilities for palliative care, the issues that affect the delivery of palliative care to this group, availability of resources, and research activity in this area	Literature review 1995–2005. 45 documents included	Not specified	Not specified	18 years or older was included in the review.	The most significant gap in research evidence is the almost total lack of insight into the needs and experiences of terminal illness from the perspective of people with intellectual disabilities themselves. If we are to provide end of life support that is appropriate and relevant to their needs, it is crucial to understand what they value in end of life care, what interventions and service provision are helpful, and where they would like to be cared for. The impact of terminal illness on the person’s social network gets minimal attention, with only a few papers making reference to this. Further studies into the palliative care needs of people with intellectual disabilities are only beneficial if steps are undertaken to act on the results.	33/39
Hanna et al. [21] Northern Ireland/UK	2011	Examine how care staff engaged in cancer prevention and health promotion activities with people with ID	An exploratory descriptive study using a postal survey design employing an anonymised questionnaire with 40 staff completed the survey on behalf of 90 adults with ID	Reports varied with how staff engaged with people with ID regarding stomach, breast, cervical and testicular cancer health promotion activities and cancer screening opportunities	Not Specified	40 staff were caring for 90 adults with ID	Health promotion and cancer prevention activities for people with ID may be less than optimal. The importance of staff training in order to raise knowledge and awareness is highlighted. Educating both staff and people with ID about the early signs and symptoms of cancer and the importance of a healthy lifestyle as a protective factor may help lead to more informed healthier lifestyle choices and lower cancer risk and morbidity.	32/39
Willis et al. [37] Scotland/UK	2015	Explores the views and experiences of paid- and family-carers when supporting women with intellectual disabilities through breast screening.	Ethnographic approach- observation and interviews- One-to-one semi-structured interviews with 13 carers (10 paid-carers, three family-carers) were undertaken and supported by periods of focused observation on behaviour related to breast awareness and breast screening.	Breast Screening	Not specified	Not Specified	Findings indicated that most women with intellectual disabilities needed some support but the quality and quantity of support depended upon both the woman’s level of intellectual disability and who was supporting them. In terms of breast screening, the findings suggested that the women were potentially being let down at all the different stages of the breast screening process, from the arrival of the invitation letter to the experience of having a mammogram. The conclusion drawn was that there was evidence of equality of service provision but inequality of service delivery and uptake. For the woman to have a good experience during the procedure there needs to be joint working by the mammographers and carers. As observed in the post- observation discussions, there were clearly mixed feelings about the roles carers and mammographers play during the procedure.	29/39
Cobigo et al., [39] Canada	2013	Review and compare cancer screening utilisation by women with intellectual and developmental disabilities (IDD) in Ontario Canada compared to other women in Ontario	The study was conducted using health administrate- databases and registries in Ontario, Canada administrative datasets (*n* = 17,777). Two cohorts were created: a cohort of all women identified as having an IDD and a cohort consisting of a random sample of 20 % of the women without (*n* = 1440,962).	Cervical and breast cancer screening	Not specified	To examine cervical cancer screening, women in the IDD and non-IDD cohorts who were 20–69 years of age on 1 April 2009 were included.	Interventions tailored to the communication skills of women with IDD are required in order to increase their knowledge of the procedure and its benefits, decrease their anxiety, and thus allow them to give informed consent. The role of family caregivers and paid staff is crucial in providing information on cervical and breast cancer screening, supporting the person during the procedure and reporting any potential symptoms of cancer to health professionals. Information and training are required to support caregivers and staff in this role. Cancer screening initiatives need to specifically consider vulnerable populations such as women with IDD when planning their strategies such as one-to-one counselling. Training and information must be provided to healthcare professionals on the importance of health screening in persons with IDD and on how to support their patients in understanding and consenting to the procedure study.	34/39
Abells et al. [40] Canada	2016	Review of the literature and provide expert opinion relating to gynaecological issues for women with developmental disabilities to support healthcare providers better understand and care for this population.	Literature review and expert opinion from the authors	Cancer screening and prevention	No details on age-inclusionWomen with developmental disabilities	Not specified	Barriers to cancer screening include lack of understanding about developmental disabilities by providers performing screening or inability to complete screening due to perceived lack of cooperation, physical barriers, patient geographical or social isolation, or lack of patient knowledge. Education and resources are needed to support patients, their caregivers, and healthcare providers to improve the overall quality of life for women with developmental disabilities and their caregivers.	33/39
Duc et al. [41] Australia	2017	This study describes the complex care of children with life-limiting conditions and intellectual disability by means of a literature synthesis and commentary with “best-practice” guide.	As so few articles were identified by formal systematic review (to be summarized in a separate study), the present authors used an expert consensus group to reference seminal paediatric palliative care papers to highlight some of the unique challenges encountered in the care of children with a life-limiting condition in the context of intellectual disability, and the urgent need for further research in this important field.	All cancers related to children with ID	Defined as “individuals with intellectual disability may have significant difficulties in both adaptive and intellectual functioning impacting communication, learning, reasoning and problem solving”.	Focus is on children, but ages not stated specifically.	Psychosocial support for the child, siblings and parents/key caregivers is of vital importance. Key priorities in caring for these children include pain and symptom management, clear communication and support for families, as they balance hope and grief with uncertainty. This is best done by providing consistent and longitudinal child-and family-centred care to children and their families—including after the death of the child or young person. Research is needed in the areas of symptom management and care coordination, communication and decision making, psychosocial and bereavement support and education/training. Children’s understanding of death and dying is another suggested focus of further research. An individualized, child-and family-centred approach is imperative—acknowledging the cultural, spiritual and emotional needs of the family and child in their home and community context, and the experiences and expertise of the child’s parents/primary caregivers.	36/39

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
