# Peer review of "What Are the Psycho-Social and Information Needs of Adolescents and Young Adults Cancer Care Consumers with Intellectual Disability? A Systematic Review of Evidence with Recommendations for Future Research and Practice"

_children, 2021, doi:10.3390/children8121118_

Round 1

Reviewer 1 Report

Thank you for tackling a timely and important topic. I happen to know this literature (well, really lack thereof) pretty well and am glad you have built on the 2020 review to include the needs of AYA with disabilities across the family cancer continuum. Most of my feedback revolves around areas where writing clarity is lacking or additional details are needed. 

First the abstract reads quite clunky because there are so many sentences about the results and implications, but not enough about the outcomes/parameters you assessed across the articles. See an example from Boles and Jones 2021 in Palliative Medicine. 

The introduction also needs.streamlining - every time I think I know what the paper will be about, the literature review changes directions. I think you start with describing cancer incidence and impacts on individual and family psychosocial functioning, then connect to identified information and supports needs of typically developing AYA patients and families, and then move into the little that has been done to consider how these experiences and needs made be different for AYA with intellectual disabilities. Then that positions your purpose statement well. 

Your purpose statement doesn't match the format used in high level systematic reviews - again see Boles and Jones 2021 for an example. It needs to be more precise to justify why you chose the inclusion and exclusion parameters that you did. 

The methodology you pulled from Green 2006 is weak - see Boles and Jones 2021 for an example of how to include qualitative research into a systematic review. 

The inclusion criteria are clear but I would remove "any study design and methodology"

There shouldn't be a "participants" section in this type of review article. 

The exclusion criteria are very unclear, especially the first three. The fifth is not needed since English language was already an inclusion criteria. 

There are inconsistencies in the language throughout the introduction and results section. Sometimes it is stated that no systematic reviews have been done regarding this topic to date, but in truth there have been systematic reviews just not as broad as this particular one. It's important to revise for consistent language that is not contradictory.

There's currently no description of how the thematic analysis was conducted. This is important since the majority of the results were framed through this thematic analysis lens. It will be important for this to be discussed explicitly in the methods section.

The last paragraph of section 4.4 abruptly changes language to speak specifically about children with life-limiting conditions. I'm confused about this transition. There should be a clearer transition statement.

The implications for practice section is incredibly clear and useful. Well done!

Author Response

On behalf of the authors, we thank you for the feedback on the review. I attach our response to both reviews. Regards, Éidín

Reviewer 2 Report

Thank you for the opportunity to review this manuscript. It is a very well conducted systematic review and I have only minor comments

The terms ‘intellectual disability’ and ‘learning disability’ seem to be used interchangeably in the text – authors should choose one term and use it consistently

Please check punctuation at the end of each sentences – often when the sentence ends with a reference there is no full stop.

Introduction – what are incidence rate estimates for cancer diagnosis in this population? How do these rates compare to rates in their non-disabled peers?

Methods – could justification for searching from 2000 onwards  be included

Results/Discussion - Authors talk about the importance of screening programs, but in reality how are these likely to be undertaken as AYA with intellectual disability do not meet age criteria for regular screening for many cancers (as far as I am aware). I assume most disease identification is likely to take place in the primary care setting? Do the authors have any recommendations about this should occur – and noting that the only tools shown to decrease unmet health need in this population are health checks – see doi.org/10.1111/dmcn.13174 and doi.org/10.3109/13668250.2015.1105939

Author Response

(The authors gave the same response as above.)

Round 2

Reviewer 1 Report

These improvements are sufficient.